# The Manifold Bioactivity and Immunoreactivity of Microbial Proteins of Cow and Human Mature Milk in Late Lactation

**DOI:** 10.3390/ani12192605

**Published:** 2022-09-28

**Authors:** Anna Maria Ogrodowczyk, Maja Jeż, Barbara Wróblewska

**Affiliations:** 1Institute of Animal Reproduction and Food Research of Polish Academy of Sciences, Department of Immunology and Food Microbiology, Tuwima 10, 10-748 Olsztyn, Poland; 2Institute of Animal Reproduction and Food Research of Polish Academy of Sciences, Department of Chemical and Physical Properties of Food, Tuwima 10, 10-748 Olsztyn, Poland

**Keywords:** cow’s milk, human milk, late phase of lactation, physiological milk microbiome, bacterial proteins, simulated digestion, immunoreactivity, antioxidant potential, bioactive peptides

## Abstract

**Simple Summary:**

The debate over the validity and benefits of breastfeeding children after the age of 1 and the superiority of human over cow’s milk is still ongoing. The recommendation of exclusive breastfeeding for about 6 months, followed by continued breastfeeding as a complementary food source for 1 year or longer, seems justified under many circumstances. The microbiological parameters of the milk play a vital role in this respect. So far, the focus has been on the qualitative profile of the microbiota, bacterial interactions with milk compounds, and the metabolites produced by bacteria. However, the role of bacterial proteins in milk, according to the authors’ knowledge, has been analyzed. It is reported that due to the disruption of the regulatory axis of the immune system in the course of hypersensitivity, organisms may give rise to decreased IgA-mediated (physiological) and increased IgE-mediated (hypersensitive) responses even to host gut microbiota proteins. In this publication, the aim was to compare whether the bacterial proteins in the mature human milk of late lactation and cow’s milk of different breeds can determine the different immunoreactive and bioactive properties of milk.

**Abstract:**

(1) Human milk (HM) is a source of many microorganisms, whose structure contains microbial protein (MP). In addition to the known health-promoting properties of HM, many activities, including immunoreactivity, may result from the presence of MP. Cow’s milk (CM)-derived MP may be 10 times more abundant than MP derived from HM. (2) Raw cow’s milk samples of Holstein and Jersey breeds, commercially available pasteurized milk, and milk from three human donors in the late lactation phase were subjected to chemical and microbiological analyzes. Microorganisms from the milk material were recovered, cultured, and their activities were tested. MPs were extracted and their immunoreactivity was tested with human high IgE pooled sera. The milk types were subjected to simulated digestion. Milk and microbial proteins were identified with LCMS and subjected to an *in silico* analysis of their activities. Their antioxidant potential was analysed with the DPPH method. (3) The MP of HM shows a stronger IgE and IgG immunoreactivity in the tests with human sera compared to the MP of CM (*p* = 0.001; *p* = 0.02, respectively). There were no significant differences between the microbes in the MP of different cattle breeds. The MS-identification and *in silico* tests of milk and microbial proteins confirmed the presence of MP with immunoreactivity and antioxidant potential. (4) MPs possess a broad bioactive effect, which was determined by an *in silico* tools. The balance between an MP’s individual properties probably determines the raw material’s safety, which undoubtedly requires further research.

## 1. Introduction

Exclusive breastfeeding is the gold standard for infant nutrition due to the multiple benefits that human milk (HM) brings to the health of the developing infants [1]. A myriad of biologically active components of HM confers protection against respiratory and gastrointestinal infections and is associated with a reduced risk of inflammatory diseases such as asthma, atopy, diabetes, obesity, and inflammatory bowel syndrome [2,3,4,5,6].

As the knowledge of the components of HM and the migration of metabolites and substances between the mother’s body and her milk deepens, the debate over the validity and benefits of breastfeeding children after the age of 1 and the superiority of humans’ over cow’s milk (CM) is still ongoing [7,8]. Nevertheless, the World Health Organization’s recommendations of exclusive breastfeeding for about 6 months, followed by continued breastfeeding as a complementary food source for 1 year or longer, are justified [9].

In particular, the role of the dynamically changing composition of the HM microbiota during the breastfeeding period is an issue that requires further research in terms of its pro-health potential [10]. There is a high microbial variability in the milk material between individuals, resulting from many factors such as diet and stress, among others [11]. So far, the focus has been on the qualitative and quantitative profile of the HM microbiota, bacterial interactions with milk compounds, and the metabolites produced by bacteria [3]. In our opinion, the role of microbial proteins (MP) is still insufficiently investigated.

Regardless of the environment, every microorganism can produce, on average, 15 femtograms of conserved proteins [12]. Importantly, protein expression is strictly correlated with taxonomic affiliation but also with growth conditions. While healthy organisms react properly to consumed MP, these proteins are a potential antigen for organisms with a disturbed Th1/Th2 balance and a MyD88-ROR-γt axis defect [13,14]. In the primary structure of MP, many peptides with various biological activities can also be located, which counteract or balance unfavorable activities, e.g., peptides with anti-inflammatory, regulatory, antioxidant, and antibacterial properties.

The MP profile and abundance, despite the mortality of microorganisms and changes in the microbiota composition during individual processes (e.g., pasteurization, normalization, fermentation, and digestion in the gastrointestinal tract), depends invariably on the initial microbiological quality of the raw material [15,16]. The hygienic limits for raw mammalian milk have been scrupulously established and normalized by the European Union (Regulation (EC) No. 853/2004) and through other legislative norms [17,18]. Since, in HM, the acceptable norm for healthy people is 10^3^–10^4^ CFU/mL of non-pathogenic microorganisms, for raw cow’s milk, this value reaches 10^5^ CFU/mL, whereas for pasteurized CM, the norm in different countries ranges between 10^3^–5 × 10^4^ CFU/mL. This means that in one liter of CM that meets the microbiological standards, there can still be 1 g of MP, while in HM it can be 100 mg.

All the aspects described above for HM also apply to the milk of other mammals, including the milk of dairy cattle [19]. The bacterial richness and biodiversity of CM are positively correlated with the age of the cattle, calving, and with the day of lactation in addition to diet, health status, and season [15,20,21]. Significant differences in the CM microbiome between different cattle breeds have also been confirmed [22]. For dairy material, the microbial profile is significant from the technological point of view. Many of the MPs are enzymatic proteins that can significantly affect the technological usefulness and attractiveness of the raw material. The microbiota of the raw milk material may influence the final characteristics of the dairy products obtained from such milk [15].

Thus, the high quantity and variability of milk’s microbial-derived proteins, in our opinion, deserve a more detailed and complex examination of their impact on the immune system of consumers. In this publication, we wanted to verify the hypothesis of whether the bacterial proteins present in a late phase of lactation in human milk and in cow’s milk, collected from different cattle breeds in their middle lactation period, could determine the different immunoreactive and bioactive properties of milk. Our work aimed to test the immunoreactive properties of the bacterial proteins derived from milk microbiota.

## 2. Materials and Methods

### 2.1. Sampling

For human milk samples, experimental designs and protocols were approved by the Bioethical Committee at the Faculty of Medical Sciences of the University of Warmia and Mazury in Olsztyn (No. 2/2016). All subjects gave their informed consent for inclusion before they participated in the study. The collection of the serum samples and the overall study was conducted in accordance with the Declaration of Helsinki and the protocol was approved by the Ethics Committee of the Medical Sciences department of the University of Warmia and Mazury in Olsztyn (No. 2/2010).

Approval by the local ethical committee was not necessary for cow’s milk sampling on a commercial farm at the time of the experiment. Samples were collected by staff with skills to conceive and perform experimental procedures according to ethical and welfare guidelines applied in our experiments.

Human milk samples from N = 3 healthy women (age range 39 to 40) during their second lactation were collected (HM1-3). Mothers followed the overall diet recommended for lactating women and were in the same late phase of lactation (breastfeeding day: 400–460). During collection, mothers were asked to completely empty the breast, with the use of a breast-pump, into sterile bottles. Each individual human milk sample consisted of 3 aliquots taken at 3 different time points of the day and pooled for each woman.

For raw cow’s milk, samples of milk in the middle lactation period from healthy Holstein (CM1) and Jersey (CM2) dairy cows were commercially purchased from two farms in the Warmia and Mazury regions. Samples were collected during the routine manual milking. All cows were housed in free housing systems, fed with total mixed ration (TMR) according to standard practices during their indoor period, and milked twice daily. Animals were in the same phase (at least the 120th day) of lactation and were milked in the same season—spring (parity: 3–4; days of milking: 120–150). Six milk samples of each breed from each farm were pooled.

For pasteurized cow’s milk, which was normalized due to fat content (2%), samples (CM3) were obtained from commercially available sources. These were obtained right after production (with the longest remaining shelf life) and chilled to 4 °C. The pooled sample consisted of 2 aliquots taken from 3 consecutive batches of the product (6 aliquots).

Immediately after milk collection, all the samples were refrigerated on ice and transported to the laboratory within an hour. Aliquots for microbiological analysis were cryopreserved in 30% (*v*/*v*) glycerol; for proteomic analysis, the aliquots were taken with a sterile syringe into low-protein-binding tubes (Z666491; Eppendorf, Hamburg, Germany), freeze-dried, and then stored at −80 °C until analysis was performed.

### 2.2. Experimental Design

A schematic presentation of the experimental design is shown in Figure 1. Initial milk samples were submitted for chemical and microbiological analyses. Microorganisms were isolated from the initial material (point 1), which were then subjected to tests for the characteristics of viability, culture, and isolation of MP (point 2). MP immunoreactivity was analyzed with human sera (point 3). The milk types were submitted to simulated gastrointestinal digestion (point 4) and protein identification (point 5) followed by *in silico* activities analyses (point 6). Both raw and digested materials were submitted to antioxidant activity assessment (point 7).

### 2.3. Chemical Analysis of Milk Samples

The chemical composition of the materials was tested with a MilkoScan™ FT2 infrared milk analyzer (Foss, Hilleroed, Denmark) calibrated to the particular material. It was used for testing 3 aliquots of each sample.

### 2.4. Simulated Digestion of Milk Samples

Milk aliquots were subjected to standardized static in vitro digestion [23]. Briefly, gastric digestion was carried out with porcine pepsin (2000 U/mg protein) (P7000; Sigma, Poznań, Poland) for 1 h at pH 3 (adjusted with 0.1 M HCl). This was followed by intestinal digestion with pancreatin (100 U/mL) (P7545; Sigma) and porcine bile salt (48305; Sigma, Poznań, Poland) to 10 mM in the final mixture at pH adjusted to 7.0 with 0.1 M NaHCO_3_, with constant stirring for 2 h at 37 °C. Samples after the intestinal step were submitted to dialysis through a 15 kDa molecular mass cut-off dialysis membrane (Spectra/Por 2.1, Spectrum Medical, Gardena, CA). All other reagents except enzymes were obtained from POCH (Poland). After digestion, the samples were immediately collected and stored at −20 °C for further analyses. Aliquots for spectral analysis were freeze-dried.

### 2.5. Microbiological Analysis of Milk Samples

Samples were thawed and homogenized prior to diluting and plating. Ten-fold dilution series were prepared in a sterile diluent [1% peptone (Difco, Lawrence, KS, USA), 0.9% NaCl, and pH 7.0] and surface-plated onto each of the following media: (a) Plate Count Agar (PCA) (105463; Merck, Warszawa, Poland), incubated at 37 °C for 48 h for enumeration of the aerobic mesophilic bacteria; (b) De Man, Rogosa, and Sharpe agar (MRS) (PS60; BIOCORP, Warszawa, Poland) incubated anaerobically for 48 h at 37 °C for enumeration of lactobacilli; (c) M17 agar (PS108; BIOCORP), incubated anaerobically for 48 h at 37 °C for enumeration of lactococci; (d) Sabouraud Dextrose Agar (SDA) (CM0041 Oxoid, Wesel, Germany) incubated for 72 h at 25 °C for enumeration of yeast; (e) Garche’s medium [24] with bacto peptone (S-0009; BTL, Łódź, Poland), incubated anaerobically for 48 h at 37 °C for enumeration of *Bifidobacterium*; (f) Tryptone Bile Agar (TBX) (CM0945; Oxoid), incubated for 20–24 h at 44 °C for enumeration of coliforms; (g) Differential Clostridial agar (DCA) (110259; Merck, Poland), incubated anaerobically for 72 h at 30 °C for the enumeration of spores of sulfite-reducing clostridia; and (h) Kanamycin aesculin azide (KAA) agar (CM0591B; Oxoid) incubated aerobically for 24 h at 37 °C for the enumeration of enterococci. The number of *Lactobacillus* and *Bifidobacterium* and spore-forming microorganisms was confirmed by morphological analysis of cells performed microscopically in phase contrast. Anaerobic conditions were maintained using the AnaeroGen™ atmosphere generation system (AN0025A; Oxoid, Thermo Scientific, Warsaw, Poland). The cell number was expressed as log CFU (colony-forming units) per milliliter.

### 2.6. Direct Recovery of Microorganisms from Milk and Their Metabolic Activity Assessment

The bacteria present in the milk samples were recovered using Brewster and Pault’s (2016) method [25]. Briefly, milk samples were centrifuged at room temperature (RT) for 15 min at 5000× *g* in a fixed-angle rotor. Samples were chilled at 4 °C, the fat layer was transferred to a fresh tube with a sterile spatula, and the supernatant was removed by aspiration. Residual fat was removed from the tube walls with a sterile swab moistened with ethanol. From both fat and milk supernatant bacterial pellets were recovered with the following protocol. The fat pellet was resuspended to the initial sample volume in PBS containing 0.1% Tween-20 (822184; Sigma, Poznań, Poland). Milk was not diluted. Both fractions were centrifuged at room temperature (RT) for 15 min at 5000× *g* in a fixed-angle rotor. Bacterial pellets sedimenting at the bottom of the test tubes were pooled from fat and milk fraction, but independently for each milk type, and washed twice with PBS (pH 7.4). The enumeration of cells with a hemocytometer preceded the preparation of suspensions for staining and culturing. 

The quantity of bacterial cells for cultures was calculated while accounting for the cell survivability after isolation, tested with fluorescent staining. A total of 1 × 10^6^ cells/mL bacterial suspensions were subjected to double staining with 50 µM 5(6)-carboxyfluorescein diacetate N-succinimidyl ester (21888; Sigma) CFDA (at 37 °C for 30 min and 30 µM propidium iodide PI (P4170; Sigma) solution and fixed with 4% Paraformaldehyde (PFA) [26]. Fluorescent intensities of CFDA (λexc = 495 nm and λem = 519 nm) and PI (λexc = 535 nm and λem = 617 nm) were measured by flow cytometry with the use of a BD LSR Fortessa Cell Analyzer (BD Biosciences, Franklin Lakes, NJ, USA) equipped with BD FACS Diva™ v.6.1 software. At least 50,000 events per sample were analyzed. Three cellular subpopulations were evaluated: viable cells (CFDA+/PI-), cell that were damaged but still metabolically active (CFDA+/PI+), and dead cells (CFDA-/PI+). Thermally inactivated cells were treated as the negative control. The results were confirmed using an epifluorescence microscope (Olympus U-RFL-T).

### 2.7. Microbial Cultures and Protein Isolation

A total of 10 ml of aliquots of MRS broth (69966; Sigma) were inoculated with 100 μL of 1 × 10^9^ CFU/mL of recovered bacteria and incubated aerobically for 18 h at 37 °C. Enumeration of bacteria and yeast in these cultures were performed as mentioned in the Section 2.4 and Section 2.5.

The whole-cell extracts of soluble bacterial proteins were obtained according to the modified Klaassens et al. 2007 protocol [27,28]. Briefly, bacterial cultures were centrifuged at RT for 15 min at 5000× *g* in a fixed-angle rotor and washed twice with PBS (pH 7.4). Bacterial cells were dispersed in isoelectric focusing (IEF) buffer containing 7 M urea (U6504; Sigma) with 2% CHAPS (3-[(3-cholamidopropyl)-dimethylammonio]-1-propanesulfonate) (C9426; Sigma), 0.65 mM of dithiothreitol (DTT) (D0632; Sigma), 0.5% IPG-Buffer, pH 3–10 (GE17-6000-87; Merck), and with PMSF protease inhibitor (S8820; Sigma). Cells were submitted to three treatments of 45 s of bead beating with a FastPrep (MP Biomedicals, ABO, Gdańsk, Poland), interspersed with 1 min on ice for cooling.

### 2.8. Protein Visualization

SDS-PAGE and Tricine-SDS-PAGE electrophoresis of milk, bacterial proteins, and hydrolysates after simulated digestion were performed according to the protocols of Laemmli (1970) and Schägger and von Jagow (1987) [29,30]. Briefly, for SDS-PAGE 12.5%, polyacrylamide gel was used; for Tricine-SDS-PAGE, the separating gel contained acrylamide and bisacrylamide at a 16.5:3 ratio, and the stacking gel at a ratio of 4:3 was made of acrylamide/bis-acrylamide solution (A3574; Sigma). Separation was conducted in a Mini PROTEAN3 Cell apparatus (Bio-Rad, Warsaw, Poland). The buffer in the Tricine method differed with the addition of Tricine to the Tris–buffer (39468; Sigma). Separation was conducted at 30 V for 30 min and 100 V for 120 min. A molecular weight marker Odyssey^®^ MW ranging from 10–250 kDa (928–40,000, Li-COR Biotechnology, Bad Homburg, Germany) was used, and the process was followed by Coomassie Brilliant Blue (R-250; Serva, Heidelberg, Germany) staining and visualization in a ChemiDoc Imaging System (Bio-Rad, Warsaw, Poland) equipped with the Image Lab (Bio-Rad) software.

### 2.9. In Vitro Assessment of Protein Immunoreactivity with Human Sera

Simultaneous detection of bacterial proteins reacting with total human IgG and IgE was performed according to Markiewicz et al.’s optimized protocol [31]. Briefly, 20 µg of proteins was SDS-PAGE-separated and transferred onto Immobilon PVDF membranes (Millipore, Warsaw, Poland). Membranes were washed in phosphate-buffered saline (PBS, pH 7.4, 5 min, RT), then blocked in the Odyssey^®^ Blocking Buffer (pH 7.2–7.6, 2 h, room temperature; 927-40003, Li-COR Biotechnology), and incubated overnight (4 °C) in a solution of human poly-allergic sera, diluted twice in blocking buffer containing 0.1% Tween 20. The characteristics of poly-allergic sera are presented in the Appendix A.

The application of human sera was approved by the Local Ethics Committee of the Faculty of Medical Sciences at the University of Warmia and Mazury in Olsztyn (No. 2/2010). After incubation with sera, membranes were rinsed four times with PBS-T buffer (PBS, pH 7.4 with 20% Tween 20). After the incubation with sera, two-color visualization was carried out through membranes incubation (90 min, RT) in a solution of two secondary antibodies: goat anti-human IgG antibodies conjugated with IRDye^®^ 800 CW (926-32232, Li-COR Biotechnology) for the detection in the green channel and mouse monoclonal anti-human IgE antibodies (I6510, Sigma) labeled using the IRDye^®^ 680 RD Protein Labelling Kit (928-38072, Li-COR Biotechnology) for the detection in the red channel. The anti-human IgG and anti-human IgE secondary antibodies were diluted at 1:15,000 and 1:500, respectively, with Odyssey^®^ Blocking Buffer (pH 7.2–7.6) containing 0.1% Tween 20 and 0.01% SDS. Signal detection was carried out with a ChemiDoc Imaging System equipped with the Image Lab software.

### 2.10. Mass Spectral Analysis of Proteins

Freeze-dried samples of milk and peptides after digestion were dissolved in 100 μL of 100 mM ammonium bicarbonate buffer, reduced in 100 mM S-Methyl methanethiosulfonate for 30 min at 57 °C, alkylated in 100 mM Tris (2-carboxyethyl) phosphine hydrochloride for 40 min, and digested overnight with 10 ng/mL trypsin (Promega, Madison, WI, USA) at 37 °C. Finally, trifluoroacetic acid (808260; Sigma) was added at a final concentration of 0.1%. Mass spectrometry (MS) analysis was performed by liquid chromatography-mass spectrometry (LC-MS/MS) in the Laboratory of Mass Spectrometry (IBB PAS, Poland) using a nanoAcquity ultra-performance LC (UPLC) system (Waters, Milford Sound, MA, USA) coupled to an LTQ-Orbitrap Velos mass spectrometer (Thermo Scientific, Waltham, MA, USA). Peptides were separated by a 180 min linear gradient of 95% solution A (0.1% (*v*/*v*) formic acid in water) to 35% solution B (acetonitrile with 0.1% formic acid) on the nano/’’ACQUITY UPLC BEH C18 Column (Waters, 75 μm inner diameter; 250 mm). Eluted peptides were directly electro-sprayed into the mass spectrometer operating in positive ion mode with a voltage of 2 kV. Spectra were recorded in full MS mode in profile mode at a resolution of 60,000 with a 400–2000 m/z scan range.

The measurement of each sample was preceded by three washing runs to avoid cross-contamination; the final LC-MS washing run was searched for the presence of cross-contamination between samples. Raw data were searched by the MASCOT server (Matrix science, London, UK) against the SwissProt database, tax.: Mammals (556,006 sequences) for milk proteins and Bacteria (Eubacteria) (114,425,735 sequences) for bacterial proteins. Search parameters were as follows: unspecific digest (enzyme—none) due to simulated digestion pretreatment in hydrolyzed samples and peptides’ mass tolerance 30 ppm, fragment ion tolerance 0.1 Da, Higher-energy collisional dissociation (HCD) with a normalized collision energy value of 35%; variable modification—methionine oxidation, methylation. The score threshold cut-off- calculated for this analysis was 43.

### 2.11. Antioxidant Activity with DPPH Radical-Scavenging Assay

The antioxidant activity of milk types and their digested samples was determined in vitro by testing their ability to scavenge 1,1-diphenyl-2-picrylhydrazyl (DPPH) radical. The method of Picot et al. (2010) [32] with modifications from Espejo-Carpio et al. [33] was used. Briefly, 1 mL of each sample (supernatant after centrifugation (14,000 rpm, 4 °C,15 min) with different protein concentrations (0.5–6 mg/mL) was added to 1 mL of 0.1 mM DPPH in methanol. The mixture was shaken and left for 30 min at RT in the dark. Finally, it was centrifuged at 10,000× *g* for 5 min, and the absorbance of the reaction mixture was measured at 517 nm (spectrophotometer; ASYS UVM 340). A blank was run in the same way by using distilled water instead of the sample, and a sample control was also made for each sample by adding methanol instead of DPPH solution. Triplicate measurements were carried out for each sample. The scavenging activity (%) was calculated as follows: scavenging activity (%) = (1 − ([absorbance of the sample − absorbance of the control]/absorbance of the blank)) * 100.

The IC_50_ value, which defines the concentration of substrate (mg protein/mL) needed to inhibit DPPH activity by half, was also calculated.

### 2.12. In Silico Protein Immunoreactivity and Antioxidant Activity Analysis

Sequences identified in LC-MS/MS proteins were retrieved from the NCBI database in FASTA format and used for *in silico* analyses. The *in silico* sequential analysis previously reported by Ogrodowczyk et al. [34] was used. Briefly, proteins were mapped to their pro-allergenic properties with online tool to confirmed allergens databases (1) AllergenOnline V.21 (http://www.allergenonline.org—last update 14 February 2021), Allergome (https://www.allergome.org/index.php—last update 10 June 2022), and WHO/IUISAllergen Nomenclature (http://www.allergen.org); (2) sequences not deposited in approved allergen databases but with tested sensitization potential in web tools, namely, Allermatch^TM^ http://www.allermatch.org (last update 31 March 2021) and AlgPred 2.0 https://webs.iiitd.edu.in/raghava/algpred2/index.html (last update 17 November 2020) were screened to find IgE epitopes. In addition, Proinflam tool for analysis and prediction of proinflammatory response of protein antigen was used (http://metagenomics.iiserb.ac.in/proinflam/index.html) (accessed on 1 June 2022). All the proteins were also tested for binding to human major histocompatibility complex class II (MHC II) with EpiTOP3 (http://www.ddg-pharmfac.net/EpiTOP3) (accessed on 31 March 2021) based on proteo-chemometric models [35] and later the IgPred tool (https://webs.iiitd.edu.in/raghava/igpred/index.html) (accessed on 31 March 2021) was used to predict different types of B-cell epitopes in tested proteins that can induce different classes of antibodies such as IgG, IgE, and IgA [27]. IL10pred tool (http://crdd.osdd.net/raghava/IL-10pred) (accessed on 31 March 2021) was used to test MHC II binders in terms of their potential to induce cytokine secretion [36]. Default settings were adopted in mentioned tools with SVM threshold of 0.2 and overlapping peptides/epitopes window length of 9 mer, as it is the shortest unit capable of MHC II activation. Finally, the BIOPEP database of bioactive peptides of food origin (https://biochemia.uwm.edu.pl/biopep-uwm/) (accessed on 1 June 2022) was used to determination of bioactive peptides with antioxidant, anti-inflammatory, antibacterial, immune-modulating, immune-stimulating, regulating, and neuroactive properties [37]. Results were presented as the frequency of bioactive fragments’ occurrence in a protein sequence (A) calculated as A = a/N, where a—the number of fragments with a given activity and N—the number of amino acid residues. The *in silico* digestion modules in online tools mimicked conditions used during the in vitro digestion. Two-step gastrointestinal algorithm pepsin > 2 pH/trypsin option was used.

### 2.13. Statistical Analysis

The normality of the variables was verified with the Shapiro–Wilk test and homoscedasticity through the Brown–Forsythe test. For chemical composition and microbiological basic comparison and summary bioactivity, parametric Student’s *t*-test was used for comparison between two analytes (cows’ vs. human milk), while for the comparison of more than two groups (accounting for the individual types of milk, the breed, and individual donor) analysis of variance was used. Data that met requirements for parametric tests were analyzed with one-way ANOVA followed by the post hoc Duncan test. All other data were analyzed with a nonparametric Kruskal–Wallis test applying module of multiple comparison tests. All calculations were performed with Statistica v. 13.1 (Statsoft, Kraków, Poland) and GraphPad Prism 9. Differences were considered significant at *p* ≤ 0.05. Data are expressed as mean ± SD.

## 3. Results

### 3.1. Chemical Composition of Milk

The chemical composition and functional properties of the milk samples are presented in Table 1. The results show that mean values of the parameters for cows’ and human milk at a late stage of lactation significantly differ in terms of protein, lactose, and dry mass content and active acidity (*p*
*<* 0.0001, *p* = 0.0005, *p* = 0.0224, and *p* = 0.0003, respectively).

In the milk from individual human donors, the content of proteins, dry mass, and urea was significantly lower than in all the milk samples from the different breeds of cows whereas the concentration of lactose was significantly higher. The richest in ingredients were the samples obtained from Jersey dairy cows (CM2) and the samples collected from human donor no. 3. In all the analyzed samples, the number of somatic cells also differed significantly; however, it did not exceed the standard adopted for materials [17]. The standards adopted for healthy human individuals are in the range of 2 × 10^3^ to 1.4 × 10^4^ cells/mL, while for raw cow’s milk that has not undergone heat treatment within 36 h from collection that number should not exceed 3 × 10^5^ cells in 1 mL, whereas after heat treatment that number should not exceed 10^5^ cells/mL.

### 3.2. Microbiological Analysis of Milk Samples

The compositions of the culturable microorganisms from the milk samples are presented in Table 2. The significant discrepancies between both types of raw materials were stated only for coliforms. In the samples of raw cow’s milk (CM1 and CM2), the abundance of CFU was two times higher than in the samples of pasteurized milk and those of human milk. The pasteurization process significantly reduced the viability of microorganisms in the milk samples—for some groups of microorganisms, up to 78%. There were not many significant differences in the quantities of culturable microbiota in the milk from cows of different breeds. The only difference was observed for Enterococci, even though the cows were at a similar stage of lactation and received the same feed. Each human sample consisted of three independent aliquots.

### 3.3. Microorganisms’ Activity after Recovery

Although not all microbes are culturable and some cannot grow after recovery from dairy material, that does not mean they are dead. This is indicated by the results of the microscopic counting of the cells and the analysis of the metabolic activity of the bacteria presented in Figure 2. At the same time, the metabolic activity was tested since even unculturable microorganisms are a source of MP. An example of this phenomenon is CM3 after the pasteurization process. For CM1, an amount of 6.1 log cells/mL was determined and for CM2 it was 5.7 log cells/mL, while for CM3 an amount of 3.8 log cell/mL was determined. Although the ratio of microorganisms capable of growth (Table 2) meets the recommended microbiological norms, the ratio of metabolically active but damaged cells remained high (CFDA^+^/PI^+^ = 46%). For all other types of raw materials, the average rate of CFDA^+^/PI^+^ was about 25%. In the raw cow’s milk samples (CM1 and CM2), this coefficiency was lower than in the human samples and it did not exceed 20%. The low dead cell ratio (CFDA^−^/PI^+^ = 3.5%) in the sample isolated from the material after pasteurization may result from the fact that the milk has been normalized in terms of fat, which is also associated with the process of the centrifugation of the sample during which the dead cells may have been eliminated with the fat. This phenomenon can also be related to the thermal processing of the milk, which deactivates cells but does not kill them.

### 3.4. Microbial Composition of Cultures of Recovered Bacteria

For the isolation of MP, cultures of culturable microorganisms were used, so it was important to use and maintain proportions of microorganisms that were representative of the individual raw material. The same level of inoculation—namely, 1% (*v*/*v*) of 100 μL of 1 × 10^9^ CFU/mL—of recovered (converted to cells capable of growth) bacteria was always applied. The proportions observed in the isolates for most of the raw materials were maintained, but in the case of the samples obtained from human donors (HM1 and 3), the share of sulfite-reducing bacteria, coliforms, and *Lactobacillus* bacteria increased significantly (*p* < 0.005, *p* < 0.05, and *p* < 0.01, respectively) at the expense of a decrease in the amount of *Lactococcus* (*p* < 0.001). However, a similar phenomenon of reducing the share of the *Lactococcus* genus was observed proportionally in all cultures, which is presented in Figure 3.

### 3.5. Characteristics of Protein Isolates

The proteins isolated from the microbial cultures were separated by SDS-PAGE and the results are shown in Figure 4a. There were differences in the band profiles between the extracts of the bacterial cultures. The greatest differentiation in the profile was observed between proteins isolated from the bacterial cultures from human donors (HM1-HM3). In these profiles, there were 29 to 38 bands. The most similar protein profiles were from the cultures of bacteria isolated from the milk samples of the two breeds of cows (CM1 and CM2), which consisted of 26 bands of the same molecular weight.

The profile of the MPs isolated from cultures of milk bacteria differed significantly from that of dairy proteins. The profile of the dairy material before and after simulated digestion is shown for comparison in Appendix A in the Appendix A. The proteins and peptides profile after digestion was dominated by whey- and casein-originated proteins and enzyme residues in the digestive profiles. However, the mass spectrometric analysis confirmed the presence of both dairy and MP in the digested milk, which is presented in Table 3 and Appendix A in the Appendix A. In the cow’s milk samples, 11 proteins of a *Bos taurus* origin dominated, including casein (alpha-S1-casein: P02662, beta-casein: P02666, and kappa-casein: A0A140T8A9), whey proteins (beta-lactoglobulin: P02754 and alpha-lactalbumin: P00711), immunoglobulin chains and their receptors (Q3SYR8; P81265), and proteins related to the bovine milk fat globule membrane (Xanthine oxidase: F1MUT3 and Perilipin: F1N1N6). Moreover, 17 microbial proteins were also identified in cow’s derived digested material, several of which (such as pyruvate kinase: C2JX14, chaperone protein DnaK: B3WEQ7, ABC transporter: C2JYM1, and enolase: D8FPZ1) were previously isolated from a beneficial *Bifidobacterium* strain but were also reported to have an immunoreactive potential [38]. The match in our study indicated the origin of these immunoreactive proteins from the *Lactobacillus* and *Lactococcus* strains. In human milk, except for 17 human milk proteins matched to *Homo sapiens* proteome (Appendix A), 17 microbial matches were also stated. Moreover, trace cow’s milk proteins with immunomodulatory potential (butyrophilin: Q13410, osteopontin: P10451-5, and lactotransferrin: E7EQB2) were identified despite the non-specific digestive process with digestive enzymes (pepsin/pancreatin). Among the immunoreactive MPs also ABC transporters: Q9CG37, R7QYF5, and E3VVN9 from *Lactococcus*, *Roseburia,* and *Lactobacillus*, and enolase: D8FPZ1 from *Lactobacillus* were present in human milk. Moreover, the immunoreactive 30S ribosomal protein S1: Q1GAK0 and 60 kDa chaperonin: A0A0E3CQW5 matched to *Lactobacillus* were detected.

### 3.6. Immunoreactivity of Microbial Proteins with Human Sera

The IgE- and IgG-immunoreactivities of the MP were tested in vitro with the sera of healthy individuals and polyallergic individuals. These results are presented in graphical form as an immunoblotting scan in Figure 4b,c, but also as a signal strength analysis in Figure 4d. The fluorescence-immunoreactive signal of human milk-derived MP indicated 2–5 times higher values, determined in arbitral units normalized to the signal of all the immunoreactive bands in the line, in comparison to the MP of cows’ milk. This was also the case for the binding of both E and G class immunoglobulines. The molecular weight range of the reactive bands (25–50 kDa) is comparable with the range of the molecular masses of the proteins identified by the MS analysis of the digests of the milk samples (ABC transporters subunits—29 kDa, 30S ribosomal protein—44 kDa, and enolase—46 kDa), while higher molecular weights are characteristic of chaperones (57–67 kDa), pyruvate kinase (62 kDa), and complete ABC transporters (66–80 kDa). As previously reported, the aforementioned proteins may be able to bind to both types of antibodies, as evidenced by the overlapping signal in our blottings.

### 3.7. Milk and Digests Antioxidant Activity

The analysis of the individual milk materials shows that there are no significant differences between the antioxidant potential of fresh samples of middle lactation period milk from the different breeds of dairy cattle and between individual donors at a late lactation phase with respect to human milk, shown in Figure 5. The values noted for the raw materials—presented in the form of the IC_50_ parameters—with a higher value indicate a lower anti-oxidative potential than for the digested material. The DPPH inhibitory activity of the raw material of two different breeds of pasteurized cow’s milk (12.5, 13.9, and 14.8%), and of the human donors’ (63.9, 52.4, and 64.7%) samples are noted, respectively. There were significant differences between the average values for the raw cow’s and human milk (*p* = 0.010) and for the fresh and pasteurized cow’s milk samples. However, the significant differences between the fresh and pasteurized cow’s milk became blurred after the digestion process. Undoubtedly, the simulated digestion process results in a significant increase in the antioxidant potential in the input raw materials, which is dictated by the release of low-molecular weight peptides < 5 kDa rich in di- and tripeptides.

The differences between the digested milk collected from individual human donors also became more significant. Invariably, in the digested material, human milk showed a significantly higher antioxidation potential; however, the significance between the average values for digested cow’s and human milk was diminished (*p* = 0.046).

### 3.8. In Silico Assessment of Bacterial Protein Immunoreactivity and Bioactive Properties

The *in silico* analysis of the properties of the identified MP is presented in Table 4. Among the MPs identified in the cow’s milk-derived cultures three proteins, namely, chaperone protein DnaK, pyruvate kinase, and enolase, show over 70% similarity with E < 1 × 10^−7^ with confirmed allergens of wheat-derived serpin—Tri a 33.0101, pyruvate kinase of striped catfish—Pan h 9, and enolase of yeast—Rho m 1, respectively. DNA damage-inducible protein and uncharacterized protein identified as *Desulfobacca acetoxidans* protein were also identified as carriers of structures homologous to the identified allergens Alt a 7 from the Alternaria plant rot fungus and Art v 1-like mugwort pollen protein. Among the microbial proteins identified in the human milk-derived material, only enolase (mentioned above) meets the criterion of possible cross-reactivity with yeast—Rho m 1 (identity/similarity over 70% and E < 1 × 10^−7^). Two other UPF0756 membrane proteins with 44% homology to Bla g 1 derived from the German cockroach and pyridine nucleotide-disulfide oxidoreductase class-II with 38% homology to Ory s 3 from rice meet the less-restrictive criterion of potential cross-reactivity.

The vast majority of MPs have binding sites for G-class antibodies. The only exception is the aforementioned UPF0756 membrane protein. Although being bound to class G antibodies is a natural phenomenon for MPs, the binding sites for class E antibodies are not well examined. Based on the primary structure of the proteins, the algorithms showed potential binding sites for class E antibodies in the following proteins: the 2,3-bisphosphoglycerate-dependent phosphoglycerate mutase and Elongation factor Tu of CM-derived bacteria and Beta-lactamase domain protein, Thymidylate kinase, ABC transporter protein, Carboxylate-amine ligase, and Chaperonin GroEL and 30S ribosomal protein S1 identified in HM-derived material. Interestingly, the presence of potential allergenic binding sites (IgE– binders) was not always consistent with the homology of the protein to the confirmed allergen. This may be due to the lack of data on the IgE-immunoreactivity of MP.

The *in silico*-determined summary number of proinflammatory epitopes for the proteins derived from microorganisms isolated from CM raw material was significantly higher (*p* = 0.047) than for the proteins isolated from cultures obtained from human donors. This estimation was also confirmed by the scores for the strongest inducer that in CM MP reached values of 2.11 for Pyruvate kinase peptides, while in MP from HM this core was 1.87 for Elongation factor G peptides.

In the structure of MP, the presence of peptides capable of binding to class A antibodies (IgA), as well as MHC II in the context of the effector activation of IL10 secretion, was also confirmed. However, no significant difference was noted between the average score values for the different raw materials (microbial CM_IgA_ = 1.17, HM_IgA_ = 1.19; CM_IL10_- 1.23; HM_IL10_-1.20).

When analyzing the presence of bioactive peptides in the individual proteins of microorganisms, the presence of numerous di and tri peptides was noted (Appendix A Excel File in Appendix A), of which 90% have antioxidant properties. The only significant difference was found for the antioxidative bioactivity of human and cow’s microbial peptides. The mean values of parameter A—characterizing the frequency of the bioactive fragments’ occurrence in a protein sequence—for the HM microbial peptides was significantly higher than for the CM microbial peptides (HMmicr_a-ox_ = 0.093; CMmicr_a-ox_ = 0.064; *p* = 0.041). For comparison, the theoretical values for all the bioactivities and antioxidant potentials of dairy peptides (results presented in Appendix A) show an opposite tendency (dairy HM_a-ox_ = 0.06; CM_a-ox_ = 0.110; *p* = 0.039). The remaining bioactive peptides of dairy material, especially the peptides of the HM proteins, also have stronger theoretical anti-inflammatory properties than the CM and MP dairy materials (HM_a-inf_ = 0.012; CM_a-inf_ = 0.008; microbial HM_a-inf_ = 0.004; CM_a-inf_ = 0.005; *p* = 0.049). In the milk raw material, the presence of a vast majority of strong cow’s milk allergens (Bos d 4–13) and their human homologs (Hom s) in HM milk have been confirmed. However, the theoretical binding intensity of dairy CM-derived proteins to class A and G antibodies was lower than for the proteins of the milk from human donors. The particular properties of the proteins of individual raw materials are, therefore, the result of many potential activities, because the same protein can be, for example, both an allergen and a source of many bioactive peptides.

## 4. Discussion

### 4.1. Microbial Proteins Immunoeactivity in Milk

For centuries, milk has been considered the most nutritious and optimal food for newborns. Regardless of the species it comes from, it is a multi-component colloidal solution rich in pro-health proteins and nutrients. It is known that human milk proteins differ in their primary structure from cow’s milk proteins [1,9,39]. That is why cow’s milk proteins, as the first cross-species protein source, are considered the main food-borne foreign antigens affecting an infant’s immune system. Among the most important components of both human and other types of mammalian milk are microorganisms [10,19,22,40]. The proteins of the microorganisms present in milk have not yet been sufficiently tested with respect to their immunoreactivity. Nonetheless, yeast proteins (*i.a.,* enolase, peroxysomal protein), with proven sensitizing potential, have been included in the new list of antigens [41,42]. Yeast is as often present in milk raw materials as other non-pathogenic microorganisms. It was also reported that some yeast heat shock proteins (Hsp) have the potential to modulate the immune response in developing newborns [43]. In that context, it has been suggested that not only milk proteins but also microbial, including bacterial proteins, may be involved in the early development of hypersensitivity and tolerance, which we reported in our previous study [28].

It is a common mistake to ignore the microbial protein content in dairy material. It is estimated that the mentioned yeasts’ protein expression reaches 6000 different proteins and analysis has revealed that around 42 million protein particles may be produced in each cell [43]. In their structures, bacterial cells also contain a large number of proteins with different functions and replication potentials. This difference in replication potential determines the number of activated protein molecules a cell can produce in different growth conditions. For example, it has been proven by other researchers that *L. casei* GCRL163 cultured under the conditions of lactose starvation showed differences in the expression of eleven glycolytic enzymes, which is a potential survival strategy under harsh growth conditions [44]. Most of the reported proteins have been observed to possess an immunoreactive potential in other studies. It was also proven that a difference in the final pH during the fermentation process (pH 4.8 and pH 5.8) causes a significant difference in the expression of 92 proteins in *Lactobacillus rhamnosus* GG and influences their phosphorylation. The phosphorylation of glycolytic enzymes was found to be especially extensive [45]. Replication potential is also dependent on the taxonomic affiliation of microbes. In Górska et al.’s 2016 study, the presence of 18 differently expressed proteins with proven immunogenic and immunoreactive potential in two human *Bifidobacterium longum* ssp. *longum* strains was confirmed [38]. The most immunoreactive proteins were identified and differed in both strains. These proteins were enolase, aspartokinase, pyruvate kinase, and DnaK in *B. longum* ssp. *Longum* CCM 7952, and sugar ABC transporter ATP-binding protein, phosphoglycerate kinase, peptidoglycan synthetase penicillin-binding protein 3, transaldolase, ribosomal proteins, and glyceraldehyde 3-phosphate dehydrogenase in *B. longum* ssp. *longum* CCDM 372. In our study, although the tested material was a mixture of MPs obtained from cultures of various bacterial genera and species, the presence of the same proteins (enolase, pyruvate kinase, chaperone DnaK, ABC transporter ATP-binding protein, and glyceraldehyde 3- phosphate dehydrogenase) with a high (>70%) degree of homology was confirmed. According to the authors of the algorithms of the accepted allergen databases AllergenOnline and WHO/IUIS Allergen Nomenclature Sub-Committee, cross-reactivity is likely for proteins with over 70% identity and similarity over the entire protein sequence. The cross-reactivity for proteins with <50% identity and similarity is less likely and requires careful verification [46]. The main aim of our study was to test the immunoreactivity of complex protein isolates representative of the milk type. That is why, instead of analyzing particular bacterial colonies, protein isolates from mixed bacterial cultures in the final phase of logarithmic growth, when protein expression is at its strongest, were analyzed. The complexity of the material, on the one hand, made the analysis difficult, but on the other hand, following identification with the use of advanced techniques of mass spectrometry and *in silico* modeling, it allowed for a comprehensive analysis of the phenomenon. The applied protocol allowed for the confirmation, with a satisfying probability, of the presence of homologs of immunoreactive bacterial proteins in cow’s raw and pasteurized milk but also in the human milk of individual donors. Therefore, the risk of cross-reactivity is possible, according to our research.

Importantly, our study also confirmed the presence of the aforementioned glyceraldehyde 3-phosphate dehydrogenase (GAPDH), which is reported to be a moonlighting protein with a huge cross-reactivity to proteins of numerous pathogens, i.e., *Staphylococcus aureus* [47], *Streptococcus agalactiae* [48], and *Edwarsiella tarda* [49]. Previously, the reactivity of GAPDH with a serum of *Clostridium difficile*-infected patients was also confirmed [38]. This phenomenon is partly explained by the observation that several proteins are strain-specific with a highly variable structure, e.g., enolase reported in *Lactobacillus* and *Bifidobacterium* strains, whereas others such as aspyruvate kinase show a high percentage of homology and share cross-reactivity [28,38,50]. It is surprising, however, that most of the proteins reported as immunoreactive are cytoplasmic proteins. So, how can the immunological system produce a humoral response to them? Importantly, many of these proteins represent moonlighting proteins and even though their main functions should be implemented inside the cell, they are exposed to the surface of bacterial cells and can often take on completely new activities.

Features such as immunoreactivity and the ability to translocate to the cell surface have been included in the moonlighting protein phenomenon and were described for the first time in 1992 for GAPDH properties [51]. Since then, it has become clear that over 100 proteins have more than one unique function and this feature is strictly related to the protein’s location [52]. All known and described moonlighting proteins have been deposited in the MoonProt database (http://www.moonlightingproteins.org) (accessed on 1 June 2022, and although further information is continually being uncovered about these proteins, there are still no exactly described mechanisms as to how bacterial moonlighting proteins translocate to the cell exterior. It is postulated that they can be passively released from dead or damaged cells or be secreted onto the cell surface in an active way [53]. Therefore, their presence, even in thermally processed milk, is entirely possible, which we have confirmed in these studies. We identified the 30S ribosomal protein and 60 kDa chaperonins (GroEL) *a.o*. that were originally cytosolic proteins, but in our study, both were the strongest immunoreactive proteins in the tested pasteurized material. The mentioned proteins were also reported to overexpress in heat-adapted lactic acid bacteria strains by other groups. For example, in *Lactobacillus plantarum* DPC2739 and DPC2741 strains treated at 72 °C for 90 s, the overexpression of 41 proteins including the ribosomal proteins L1, L11, L31, and S6; DnaK; and GroEL was confirmed [54].

Importantly, protein expression is strictly correlated not only with taxonomic affiliation and the growth conditions of microorganisms but also with their further processing. Even short-term environmental stresses, e.g., during digestion (in the presence of bile salts and a low pH), may affect the expression of microbial proteins [55,56]. A significant change was reported in the expression of 347 proteins of the *Lactobacillus salivarius* LI01 strain in response to bile stress, and 27 proteins of *L. kefiranofaciens* M1 changed expression in response to heat, cold, acid, and bile salt stresses. Interestingly, even physiological processes, e.g., those related to the autolysis of bacterial cells, can cause changes in protein expression [57]. The significant change in bacterial protein profile has been confirmed in the acid-resistant *L. paracasei* GCRL46 strain submitted to persistent acidity (pH-4). The hydrolases produced by that strain, including amidases and peptidases, play a critical role in peptidoglycan turnover during growth, thereby impacting daughter cell separation as well as cell death through autolysis. However, persistent acidity decreased natural hydrolases expression but boosted the expression of S protein, homologous to the glucan-binding (GbpB) surface antigen (SagA) proteins detected in pathogenic group A streptococci species as secreted, immunoglobulin-binding (SibA) proteins (also named PcsB) [58]. Several proteins in our study coincided with the proteins reported above.

That was the reason why, in our study, an additional step of simulated digestion was applied; however, none of the protein homologous to shock proteins and glucan-binding surface antigen was identified. The vast majority of proteins represented the most common of the commensal bacterial moonlighting proteins lacking fully understood bioactive and immunoreactive properties. Even though, over the past 25 years, they have been found to be rather widespread in bacteria, their role in immune system regulation needs further study. An example of a further unexplored niche regarding this aspect is one of the latest reports on *Bacillus subtilis’s* up-regulation of moonlighting protein activation via small proteins (such as SR1P and SR7P), DNA, RNA binding, or protein phosphorylation [59]. Haq et al.’s group also concluded that one of the two GAPDHs in *B. subtilis*, GapA, is a moonlighting protein with a metabolic function in glycolysis and a second function in RNA degradation [59].The same was stated for enolase. As both GAPDH and enolase are abundant proteins required for glycolysis in all organisms, it cannot be dismissed that it moonlights in the RNA degradation in other bacteria, too. Interestingly, in the current study’s process of modeling enolase bioactivity *in silico*, no sequence with identified immunological, antioxidant, or regulatory activity was identified, despite the relatively large (46 kDa) size of this protein. In the reported MoonProt database, it was mentioned that *Steinernema glaseri* (nematode-derived) enolase can invoke the antiphagocytic activity of the host (insect) and affect its immune suppression.

Such high dynamics of change in the expression and activation of microbial proteins are, therefore, an issue that requires greater attention not only from a health-related perspective but also in terms of the technological defects of potential products made of such raw materials containing microbial proteins. However, the safety of the raw material is a priority because it is reported that due to the disruption of the regulatory axis of the immune system in the course of hypersensitivity, organisms may give rise to decreased IgA-mediated (physiological) and increased IgE-mediated (hypersensitive) responses even to host gut microbiota proteins [13,14]. In response to these proteins, it is not a specific allergy that is important, but the phenomenon of the induced hypersensitivity pathways and the MyD88-ROR-γt axis defect, which may be congenital or secondary to dysbiosis. We conclude that because the proteins identified in the mixtures of the bacterial proteins isolated from the milk’s raw material showed the ability to bind with class E antibodies, this may be indicative of the presence of a person with a disturbance in the functioning of the mentioned regulatory axis among the serum donors. Human sera from patients with polyallergies to food products, pollen, and animal dander in addition to specific E antibodies manifested elevated total levels of IgE > 100 kU/L (above the norms for their age). With those sera, we indicated the presence of positive IgE-mediated and IgG-mediated reactions in the in vitro tests. We also confirmed this possibility during the *in silico* modeling of the epitopes of microbial proteins for particular antibody classes. In this regard, even the presence of a trace amount of microbial protein in breast milk can play a significant role in the baby’s developing immune system. Perhaps it is a factor of the natural activation of the immature immune system of children by the microbial antigens present in the milk. With the naturally decreasing number of human antibodies in human milk, the role of microbial proteins as a source of cross-species protein increases.

In our research, the cultures of microorganisms in the MRS medium were also used for the isolation of proteins of a microbial origin. The use of a medium with a low selective and differentiating potential but which was optimal for *Lactobacillus* bacteria was applied because in the milk material both from the donors and dairy cattle and in the pasteurized milk, the frequently reported ‘secondary’ microbiota are bacteria belonging to the *Lactobacillus* and *Lactococcus* genera [15,17,18,19]. In norms for cow’s raw milk, the pasteurization’s efficiency should range from 90 to 99%. Thus, the number of non-pathogenic microorganisms should not exceed 1 × 10^4^ CFU/mL (PN-90/A-86003). We confirmed the presence of active microorganisms in both the microbial samples recovered from the raw material and in the culturable microbe suspensions. In all cases, the abundance of live and active microorganisms has not exceeded the acceptable norms. The viability and activity of the bacterial cells were high. The only exceptions were the pasteurized milk samples where the raw material was previously normalized in terms of fat content and which were heat treated. During these processes, the vast majority of bacterial cells are centrifuged and trapped in the fat fraction; therefore, in our research, the aeration was carried out from the liquid and fat fractions in relation to all the raw materials in the same way. Thus, the pasteurized raw material was initially characterized by a lower number of microorganisms and the activity of these cells was significantly reduced; additionally, more material was initially required to obtain the appropriate number of cells for the inoculation and culture. Despite the proportions of the cultivated bacteria changing significantly compared to the raw milk, the biodiversity was sustained regardless of the starting material. Such a protocol allowed for the isolation of an adequate number of microbial proteins necessary for all the analyses. Nevertheless, to maintain the most faithful representation of the phenomena occurring in the milk-consuming organisms, the identification of microbial proteins was carried out from digests of dairy materials and not from bacterial isolates. This same protocol was included in the antioxidant potential study.

In the analyses of the antioxidant potential of the raw cow’s and human milk [60,61,62], the pasteurized milk [63], and the digested milk [33], no deviations from the results of other groups were noted. No matter the method of the antioxidant potential measurement used, human milk was characterized by an invariably higher potential than the other milk types. It should be remembered that various milk ingredients are responsible for the antioxidant potential of human milk, but many of these compounds are degraded during the thermal processing of the raw material, which was emphasized in Martysiak-Żurowska et al.’s [64] research. The authors tested the antioxidant potential of human milk with ABTS and TBARS tests and obtained results indicating a high pro-health potential as necessary for healthy growth at every stage of an infant’s life, but also stressed that this milk loses its value during the pasteurization process. On the other hand, Pozzo et al. [65] also discuss the health potential of human milk but mentioned that its antioxidant potential, tested with ABTS and DPPH methods, was lower than infant formulas dedicated to preterm and low birth weight infants. It was stated that the use of human milk minimizes the intake of dietary oxidative compounds in comparison to infant formulas due to the malondialdehyde content. The addition of such fortifiers to human milk increases its antioxidant capacity; additionally, the choice of the protein source (hydrolyzed vs. whole proteins) differently impacts the resulting total antioxidant capacity of the diet. Invariably, protein is one of the important components of all milk varieties.

In their primary structure, both milk proteins and microbial proteins have numerous di- and tripeptides proven antioxidant potential. Despite the many other anti-oxidant factors of milk, in this publication, we focused only on this activity with respect to proteins and discussed this activity in the context of microbial proteins and their peptides. On the one hand, enzymatic hydrolysis, with the application of conditions reflecting the digestive tract, made it difficult to search for and identify proteins during the mass spectrometry analysis due to the lack of specificity towards the method of digestion (other than trypsin); on the other hand, it allowed us to track significantly identified proteins that may appear in physiological conditions. Moreover, allowed for the same selection of conditions for the *in silico* analysis. During the analysis, the higher modeled antioxidant activity of the human milk proteins and microbial proteins isolated from human milk was confirmed. This could be a decisive factor regarding the validity and superiority of human milk, even in the later stage of lactation, over the middle lactation period of cow’s milk. Interestingly, thermal processes also favorably impacted the release of peptides with antioxidant effect, while the digestive process determined the slow blurring of the differences between human and cow’s milk. The peptides released in the digestive process comparably increased the antioxidant potential of both CM and HM.

### 4.2. Study Design Limitations and Analytical Limits

Our studies were limited by the small number of samples of human milk. However, we were focused on the unification of the tested group. It was important for us to maintain a homogenous material influenced by limited variables. Further studies on more abundant materials and other period-derived materials will be necessary.

The analysis of the antioxidant potential, due to the compounds used to extract bacterial proteins, could only be carried out on fresh and digested milk material and not in bacterial protein extracts. Compounds of extraction, isoelectric, focusing (IEF) buffer such as urea, DTT, CHAPS could influence of test results that is why we tested hydrolysates, not extracts. The *in silico* antioxidant potential analysis, however, considered both the dairy material and microorganisms’ proteins in order to at least partially reflect the antioxidant potential of the microbially derived proteins. The DPPH method—chosen for assessing antioxidant potential—as a single method is not always the ideal solution. The results obtained by us, however, do not differ from the results of other teams and were additionally supported by analysis with *in silico* modeling.

### 4.3. Plans for Further Research

As the topic of microbial protein in dairy raw material is mentioned herein for the first time in the context of its immunomodulatory potential, further research should include the analysis of the raw dairy material of dairy cattle at various stages of lactation, and the same should apply to human milk. The influence of factors such as diet, stress, the order of the offspring, or other environmental factors should also be taken into account.

## 5. Conclusions

In our study, we confirmed that bacterial proteins of culturable, milk-derived bacteria can induce humoral antibodies of E and G class-related reactions. Even though microbial proteins in milk material are not abundant, they represent a broad spectrum of bioactivities dominated by antioxidant properties. We observed that the significant microbial proteins present in all the types of milk materials are GAPDH, DnaK, GroEL, and enolase. All of these proteins are capable of inducing IgE- and IgG-mediated reactions and induce both inflammatory (IL-2 and IL-6-related) but also regulatory (IL-10-related) pathways.

Microbial proteins are an important component of milk with a significant immunogenic and bioactive effect. The balance between the individual properties of microbial proteins probably determines the safety of the raw material, which undoubtedly requires further research. Therefore, a high dynamic of change in the expression and activation of microbial proteins is an issue that requires greater attention not only from a health-related perspective but also in terms of the technological defects of potential products made of such raw materials containing microbial proteins. However, the safety of the raw material is a priority because it is reported that due to the disruption of the regulatory axis of the immune system in the course of hypersensitivity, the reaction to proteins’ commensal microorganisms is highly possible. This reaction is caused by the over 70% homology of the microbial proteins to the putative allergens, moonlighting phenomena, and an individual organism’s sensitivity. In this regard, even the presence of a trace amount of microbial protein in breast milk can play a significant role in the baby’s developing immune system. Perhaps it is a factor of the natural activation of immature immune systems of children through the microbial antigens present in the milk. The role of microbial proteins as a source of cross-species protein is important. Therefore, such a phenomenon may justify breastfeeding with the mature late lactation phase milk.

## Figures and Tables

**Figure 1 animals-12-02605-f001:**
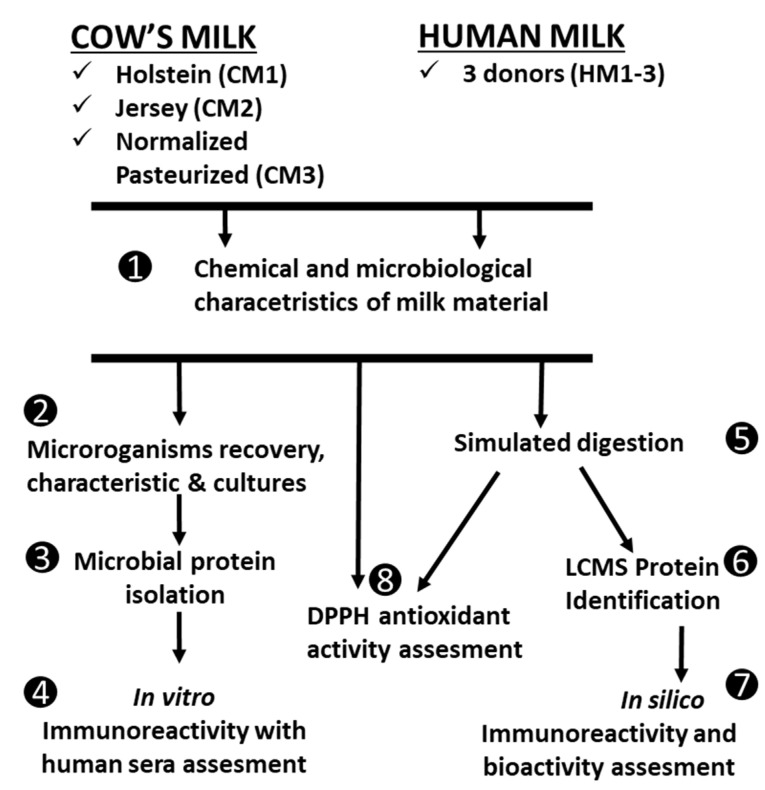
Experimental design.

**Figure 2 animals-12-02605-f002:**
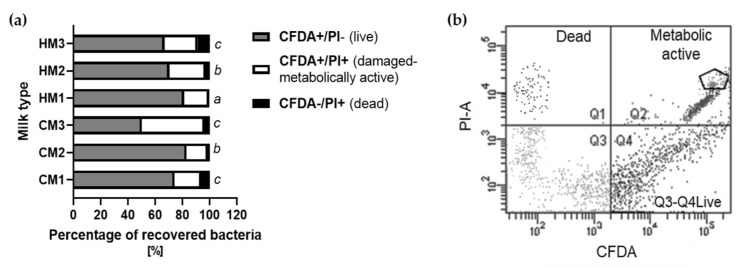
Metabolic activity of microorganisms after their direct isolation from milk samples: (**a**) percentage of microbial cells in various states after their recovery from milk samples; (**b**) exemplary dot blot with gating, including the pentagonal gate marked where the calibration bits for the cytometer are located. Different letters denote significant difference with respect to the Duncan test; *p* < 0.05 values for metabolic activity (grey + white bars) of microorganisms isolated from different materials.

**Figure 3 animals-12-02605-f003:**
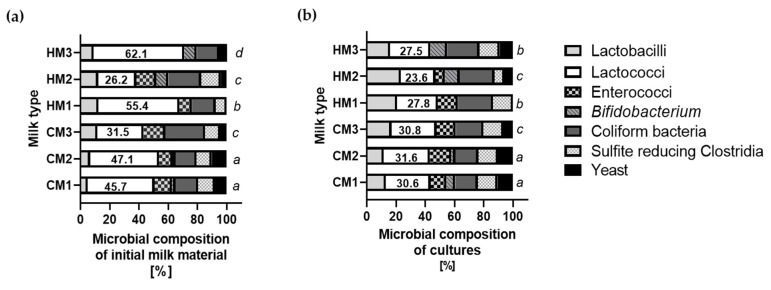
The composition of initial milk material isolated microorganisms (**a**) and microbial cultures (**b**). Different letters denote significant difference with respect to Duncan test, and *p* < 0.05 values for the proportion of microorganisms represent the compositions from different materials.

**Figure 4 animals-12-02605-f004:**
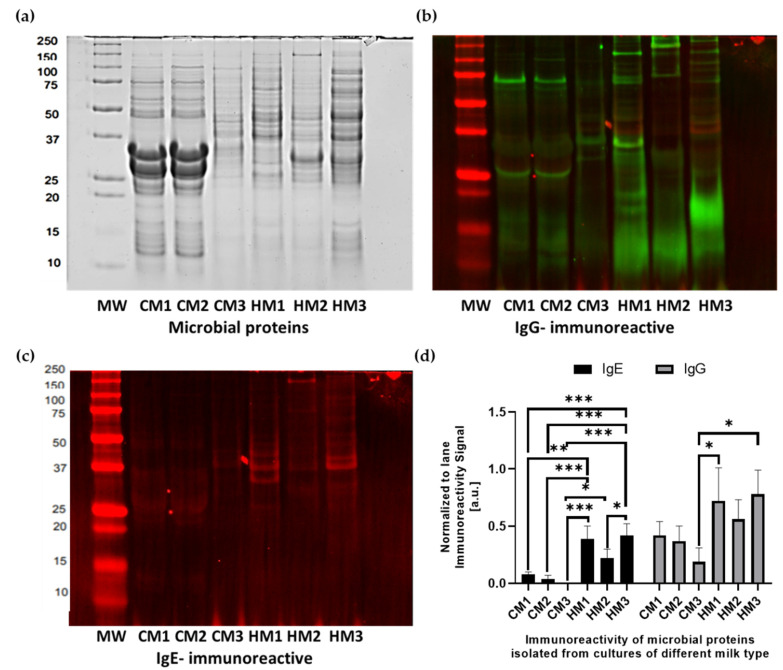
The in vitro determination of bacterial proteins’ immunoreactivity tested with human allergic serum. (**a**) SDS PAGE separation of bacterial proteins’ isolates; (**b**) IgG-immunoreactivity with IRDye^®^ 800 CW; (**c**) IgE-immunoreactivity with IRDye^®^ 680 RD; (**d**) Signal of immunoreactive bands normalized for each lane by deriving the ratio of the signals of the total protein in each lane. MW—molecular weight marker; CM1-CM3—proteins of bacteria isolated from cultures based on cows’ milk microbial isolates; HM1-HM3—proteins of bacteria isolated from cultures based on human milk microbial isolates. Asterisks denote significant difference with respect to the Duncan test values of immunoreactivity for the proteins isolated from microorganisms from different milk types.

**Figure 5 animals-12-02605-f005:**
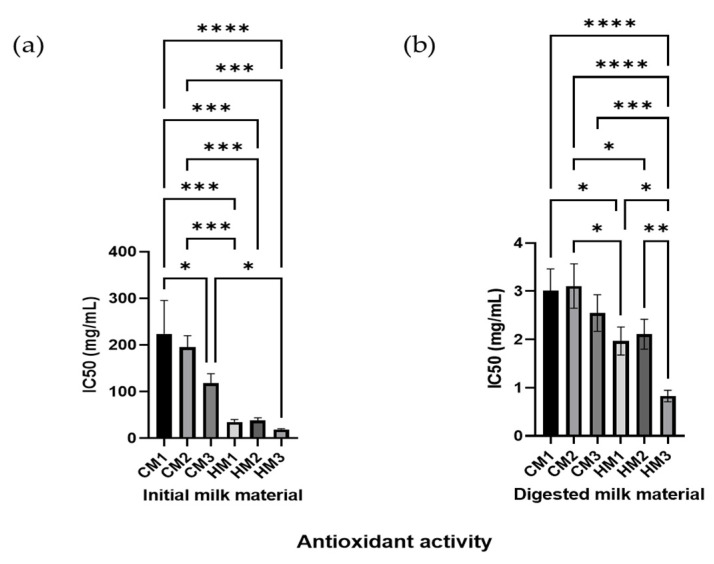
The antioxidant capacities of dairy materials and their digests. Presented as IC_50_ antioxidant activity for the different initial milk materials (**a**) and for different digests (**b**). Asterisks denote values significant difference with respect to Duncan test at * *p* < 0.05, ** *p* < 0.01, *** *p* < 0.001, or **** *p* < 0.0001.

**Table 1 animals-12-02605-t001:** Chemical composition and functional parameters of milk.

Parameters/Samples	CM1	CM2	CM3	HM1	HM2	HM3	*p* Values CM vs. HM
Proteins [%]	3.14 ± 0.31^a^	3.35 ± 0.33 ^a^	3.09 ± 0.37 ^a^	1.12 ± 0.23 ^b^	1.25 ± 0.1 ^b^	1.42 ± 0.13 ^b^	**<0.0001**
Casein [g/100 mL]	2.51 ± 0.21 ^b^	3.12 ± 0.13 ^a^	2.97 ± 0.05 ^a^	0.43 ± 0.08 ^c^	0.39 ± 0.08 ^c^	0.49 ± 0.12 ^c^	**0.0048**
Total fat [%]	3.58 ± 0.12 ^b^	4.32 ± 0.21^a^	2.92 ± 0.11 ^c^	3.91 ± 0.45 ^a^	3.6 ± 0.14 ^b^	4.81 ± 0.28 ^a^	0.4101
Dry skimmed mass [%]	9.05 ± 0.28 ^b^	9.59 ± 0.20 ^a^	9.04 ± 0.16 ^b^	7.38 ± 0.48 ^c^	8.2 ± 0.41^c^	8.21 ± 0.29 ^c^	**0.0224**
Lactose [%]	5.29 ± 0.21 ^b^	5.35 ± 0.29 ^b^	4.89 ± 0.06 ^b^	7.28 ± 0.32 ^a^	7.52 ± 0.39 ^a^	7.18 ± 0.4 ^a^	**0.0005**
Urea [mg/L] *	197 ± 16.5 ^a^	158 ± 13.5 ^b^	135 ± 10.0 ^d^	124 ± 18.2 ^e^	148 ± 10.0 ^c^	136 ± 11.5 ^d^	0.2671
**Functional parameters**							
Active acidity [°T]	19.45 ± 0.1^a^	18.71 ± 0.14 ^a^	17.63 ± 0.18 ^b^	9.81 ± 0.05 ^c^	8.75 ± 0.12 ^d^	9.92 ± 0.08 ^c^	**0.0003**
Freezing point [°C] *	0.536 ± 0.05	0.535 ± 0.07	0.496 ± 0.06	0.529 ± 0.06	0.526 ± 0.05	0.561 ± 0.06	0.3995
Somatic cells [×10^3^ cells/mL]	122 ± 1.19 ^b^	141 ± 0.95 ^a^	98 ± 0.78 ^c^	3.65 ± 1.15 ^d^	5.17 ± 0.99 ^d^	4.3 ± 1.19 ^d^	**0.0016**

Different superscript letters in the rows denote significantly different values of the Duncan (Kruskal–Wallis *) test for particular materials including breeds of cows and human donors. The *p*-values of the Student’s *t*-test for comparison of CM vs. HM were considered significantly different at *p* < 0.05 and are in bold in the last column.

**Table 2 animals-12-02605-t002:** Microbial quantity in initial milk samples.

Growth of Bacteria (log_10_ CFU/mL)	Milk Types						*p* Values CM vs. HM
	CM1	CM2	CM3	HM1	HM2	HM3	
Aerobic mesophilic bacteria	4.87 ± 0.24 ^a^	4.64 ± 0.45 ^a^	2.15 ± 1.02 ^b^	3.4 ± 0.7 ^a^	2.97 ± 0.27 ^b^	2.82 ± 0.75 ^b^	0.406
Lactobacilli	0.36 ± 0.09	0.48 ± 0.28	0.37 ± 0.18	0.54 ± 0.28	0.39 ± 0.16	0.3 ± 0.13	0.937
Lactococci	3.28 ± 0.76 ^a^	3.37 ± 0.25 ^a^	1 ± 0.35 ^c^	2.43 ± 0.42 ^b^	0.84 ± 0.6 ^c^	2.08 ± 0.08 ^b^	0.448
Enterococci *	0.85 ± 0.05 ^a^	0.67 ± 0.12 ^b^	0.47 ± 0.18 ^c^	0.37 ± 0.06 ^c^	0.43 ± 0.08 ^c^	0	0.084
*Bifidobacterium **	0.17 ± 0.68	0.13 ± 0.58	0	0	0.27 ± 0.28	0.28 ± 0.18	0.472
Coliform bacteria	1.11 ± 0.31 ^a^	1.04 ± 0.07 ^a^	0.87 ± 0.07 ^a^	0.72 ± 0.2 ^a^	0.72 ± 0.3 ^a^	0.52 ± 0.2 ^b^	**0.022**
*Escherichia coli **	0.16 ± 0.17	0.18 ± 0.03	0	0	0.13 ± 0.05 ^a^	0	0.383
Sulfite reducing clostridia *	0.84 ± 0.14 ^a^	0.74 ± 0.25 ^a^	0.33 ± 0.59 ^a^	0.33 ± 0.21 ^b^	0.43 ± 0.75 ^a^	0	0.132
Yeast *	0.56 ± 0.24 ^a^	0.72 ± 0.12 ^a^	0.13 ± 0.08 ^b^	0	0.13 ± 0.18 ^b^	0.17 ± 0.68 ^a^	0.114

Different superscript letters in the rows denote significantly different values of the Duncan (Kruskal–Wallis *) test for particular materials including breeds of cows and human donors. The *p*-values of the Student’s *t*-test for comparison. CM vs. HM were considered significantly different at *p* < 0.05 and are bolded in the last column.

**Table 3 animals-12-02605-t003:** Microbial proteins identified by LC-MS/MS analysis in milk sample digests.

Protein Name	Accession ^a^	Score ^b^	Mass ^c^	Matches ^d^	emPAI ^e^	Protein Sequence Coverage ^f^	Host Organism
**Microbial proteins from CM samples**							
Carboxylate-amine ligase	A0A0E2MAL2	278	51,305	40	0.09	9	*Lactobacillus casei*
Lipid kinase	D8FRG9	262	37,442	15	0.36	15	*Lactobacillus rhamnosus*
Transcriptional regulator	Q9CHX7	243	16,252	10	0.29	8	*Lactococcus lactis*
2,3-bisphosphoglycerate-dependent phosphoglycerate mutase	A0A0L0RK75	134	25,938	11	0.62	12	*Lactobacillus rhamnosus*
Endo-1,4-beta-xylanase	Q9CIS3	107	41,689	10	0.25	13	*Lactococcus lactis*
Pyruvate kinase	C2JX14	63	62,809	4	0.22	5	*Lactobacillus rhamnosus*
Chaperone protein DnaK	B3WEQ7	57	67,523	5	0.21	4	*Lactobacillus casei*
Putative Molybdenum transport system protein	L8JA30	57	30,581	7	0.15	6	*Photobacterium marinum*
Cell division protein	S6AB11	57	33,270	4	0.23	12	*Sulfuricella denitrificans*
ABC transporter	C2JYM1	56	66,162	4	0.29	5	*Lactobacillus rhamnosus*
DNA damage-inducible protein	A0A0F8UZN5	52	47,252	2	0.09	5	*Aspergillus rambellii*
Arginine exporter protein	K8X559	46	22,849	3	0.20	8	*Providencia burhodogranariea*
Acetyl-coenzyme A	Q1GAF3	45	31,531	3	0.14	6	*Lactobacillus delbrueckii*
Uncharacterized protein	F2NFM8	44	44,515	2	0.10	5	*Desulfobacca acetoxidans*
Elongation factor Tu	A0A1X9TWP9	44	43,330	2	0.10	5	*Lactobacillus delbrueckii*
Uncharacterized protein	Q9CFV8	44	12,459	2	0.39	9	*Lactococcus lactis*
Enolase	D8FPZ1	44	46,282	2	0.20	4	*Lactobacillus delbrueckii*
**Microbial proteins from HM samples**							
ABC transporter ABC binding and permease protein	Q9CG37	155	68,327	11	0.11	6	*Lactococcus lactis*
Beta-lactamase domain protein	B9XAB6	102	50,319	15	0.09	7	*Pedosphaera parvula*
Uncharacterized protein	C4QZG1	78	7216	2	0.75	17	*Komagataella pastoris*
Acyl_transf_3 domain-containing protein	R6MBA2	76	44,535	2	0.10	9	*Bacteroides clarus*
Cyclopropane-fatty-acyl-phospholipid synthase	Q1G9E5	68	45,321	6	0.10	8	*Lactobacillus delbrueckii*
Pyridine nucleotide-disulfide oxidoreductase class-II	A0A0D9MGW0	60	36,403	2	0.12	10	*Penicillium solitum*
ABC transporter related	R7QYF5	57	29,417	4	0.15	13	*Roseburia sp.*
Thymidylate kinase	Q9CIG4	55	23,984	2	0.38	8	*Lactococcus lactis*
Glyceraldehyde 3-phosphate dehydrogenase	K2HVL5	52	37,976	6	0.18	11	*Bifidobacterium bifidum*
ABC transporter protein	E3VVN9	50	80,255	5	0.11	2	*Lactobacillus casei*
Elongation factor G	Q1GBM0	50	76,506	4	0.12	4	*Lactobacillus delbrueckii*
Enolase	D8FPZ1	49	46,282	12	0.80	13	*Lactobacillus delbrueckii*
Carboxylate-amine ligase	A0A0U3CLC8	47	48,369	2	0.19	3	*Lactobacillus delbrueckii*
60 kDa chaperonin	A0A0E3CQW5	46	57,357	3	0.16	3	*Lactobacillus rhamnosus*
UPF0756 membrane protein	B3WE66	45	15,858	2	0.69	10	*Lactobacillus casei*
30S ribosomal protein S1	Q1GAK0	44	44,211	3	0.21	5	*Lactobacillus delbrueckii*
Lipoprotein	K0MV23	44	30,417	2	0.15	2	*Lactobacillus casei*

^a^ UniProt/Trembl database accession number. ^b^ The protein score derived from the ion scores of MS/MS report based on the calculated probability; when the significance threshold was chosen to be 0.05, the score’s cut-off was 43. ^c^ Monoisotopic mass expressed in daltons (Da). ^d^ Number of significant peptide matches. ^e^ The Exponentially Modified Protein Abundance Index (emPAI). ^f^ Coverage expressed in % of amino acids in a specific protein sequence that was found in significant peptide matches.

**Table 4 animals-12-02605-t004:** The *in silico* analysis of properties of microbial proteins identified in milk types. (**A**) In cow’s milk (CM); (**B**) In human milk (HM).

A)													
Protein Name		Accession Number ^a^	Allergenicity ^b^	Proinflamatory Response ^c^	Immunoglobulin Induction ^d^	IL10 Inducers ^e^	Bioactivity ^f^						
							Anti-Inflammatory	Antioxidative	Antibacterial	Immuno- Modulating	Immuno- Stimulating	Regulating	Neuropeptides
Microbial proteins from CM samples													
Carboxylate-amine ligase		A0A0E2MAL2	no matches	214/1.52	-	1.36	0.0045	0.0839	-	0.0068	-	0.0091	0.0023
Lipid kinase		C2JW45	28.8% identity (63.0% similar) with allergen Lat c 1; E = 0.062	152/1.46	IgG-1.365	1.28	0.0029	0.0698	-	0.0087	-	-	0.0116
Transcriptional regulator		Q9CHX7	29.0% identity with allergen Alt a 15.0101; E = 1.3	109/1.64	IgG-1.059	1.73	-	0.1167	-	-	-	0.0333	0.0167
2,3-bisphosphoglycerate-dependent phosphoglycerate mutase		C2JZH4	31% identity with Clo bo Toxin; E = 0.57	98/1.44	IgG-0.944; IgE-1.167	1.28	0.0087	0.0873	-	0.0087	-	-	0.0087
Endo-1,4-beta-xylanase		Q9CIS3	26.4% identity (63.2% similar) with allergen Sus s Laminin; E = 0.68	168/1.44	IgG-1.287; IgA-1.356	1.26	-	0.1167	-	-	-	-	-
Pyruvate kinase		C2JX14	41.9% identity (73.8% similar) with allergen Pan h 9; E < 1 ×10^−7^	266/2.11	IgG-1.182; IgA-1.114	0.96	-	0.0391	-	0.0017	0.0017	0.0119	0.0068
Chaperone protein DnaK		B3WEQ7	53.0% identity (85% similar) with allergen Tri a 33.0101; E < 1 ×10^−7^	309/204	IgG-1.359; IgA-0.998	0.72	0.0032	0.0689	-	0.0016	0.0016	0.0112	0.0032
Putative Molybdenum transport system protein		L8JA30	no matches	130/1.28	IgG-1.180	1.03	0.0036	0.0427	-	-	0.0036	0.0071	0.0071
Cell division protein		S6AB11	no matches	171/1.68	IgG-1.091; IgA-0.977	1.55	-	0.0596	-	-	-	0.0265	-
ABC transporter		C2JYM1	no matches	299/1.69	IgG-1.458; IgA-0.946	1.40	0.0034	0.0604	0.0167	0.0017	-	0.0134	0.0168
NMD3 family protein		A0A0N8HZU5	25.7% identity (52.5% similar) with allergen Tyr p 2.0101; E = 0.17	58/1.13	IgG-1.148; IgA-0.996	0.80	-	0.0568	-	-	-	0.0108	0.0081
DNA damage-inducible protein		A0A0F8UZN5	41.3% identity (60.9% similar) with allergen Alt a 7; E = 0.1	204/1.35	IgG-1.291	1.05	0.0023	0.0508	-	0.0046	0.0046	0.0185	0.0023
Ragulator complex protein		A0A0P7YKP3	no matches	22/1.13	IgG-1.599	1.38	-	0.0696	-	-	-	0.0087	0.0174
Arginine exporter protein		K8X559	no matches	72/1.11	IgG-1.211	1.05	-	0.0878	-	0.0878	0.0049	0.0146	0.0195
Acetyl-coenzyme A		Q1GAF3	28.0% identity (60.0% similar) with allergen Pen m 2; E = 0.66	125/1.38	IgG-1.345;	1.23	-	0.0567	-	0.0106	0.0035	0.0106	0.0106
Uncharacterized protein		F2NFM8	44.7% identity (57.9% similar) with Art v 1-like protein; E = 0.4	113/1.28	IgG-1.292; IgA-1.181	1.34	0.0050	0.0723	-	-	-	0.0249	0.015
Elongation factor Tu		A0A1X9TWP9	26.0% identity with allergen Der p 1; E = 2.3	139/1.43	IgG-1.226; IgE-1.152; IgA- 1.111	1.21	0.0051	0.0530	-	0.0025	-	0.0202	-
Uncharacterized protein		Q9CFV8	26.7% identity (55.8% similar) with allergen Gly m 6.0501; E = 0.95	1/0.71	IgG- 1.235	1.65	-	0.0500	-	-	0.0167	0.0167	0.0167
Enolase		D8FPZ1	50.2% identity (76.3% similar) with allergen Rho m 1; E < 1 ×10^−7^; score 0.75	155/1.21	IgG-1.105; IgA-0.988	1.03	-	-	-	-	-	-	-
**B)**													
**Protein Name**	**Accession Number ^a^**		**Allergenicity ^b^**	**Proinflamatory Response ^c^**	**Immunoglobulin Induction ^d^**	**IL10 Inducers ^e^**	**Bioactivity ^f^**						
							**Anti-Inflammatory**	**Antioxidative**	**Antibacterial**	**Immuno- Modulating**	**Immuno- Stimulating**	**Regulating**	**Neuropeptides**
**Microbial proteins from HM samples**													
ABC transporter ABC binding and permease protein	Q9CG37		no matches	292/1.7	IgG-1.314	1.5	-	0.1833	-	0.0167	-	0.0167	-
Beta-lactamase domain protein	B9XAB6		26.0% identity with allergen Eur m 4.0101; E = 1.3	186/1.25	IgG-1.356; IgE-0.941; IgA-1.096	1.54	0.0022	0.0714	-	-	0.0022	0.0325	0.0087
Acyl_transf_3 domain-containing protein	R6MBA2		27.0% identity with allergen Can f 7; E = 0.91	210/1.36	IgG-1.292	1.52	0.0104	0.1013	-	0.0026	-	0.0208	0.0208
Cyclopropane-fatty-acyl-phospholipid synthase	Q1G9E5		no matches	184/1.55	IgG-1.286; IgA-1.262	1.25	0.0102	0.0738	-	0.0051	-	0.0153	0.0102
Pyridine nucleotide-disulfide oxidoreductase class-II	A0A0D9MGW0		38.0% identity with putative allergen Ory s 3; E = 1.3	108/1.22	IgG-1.787	0.93	-	0.0950	-	0.0059	-	0.0148	0.0178
ABC transporter related	R7QYF5		no matches	104/1.48	IgG-1.304	1.44	-	0.1541	-	-	-	0.0077	0.0154
Thymidylate kinase	Q9CIG4		no matches	58/1.23	IgG-1.324; IgE-1.049	1.13	-	0.0333	-	-	-	0.0500	-
Glyceraldehyde 3-phosphate dehydrogenase	K2HVL5		41.4% identity (71.7% similar) with allergen Tri a 34.0101; E < 1 ×10^−7^	101/1.52	IgG-1.252; IgA-1.257	0.83		0.0667	-	-	-	-	0.0333
ABC transporter protein	E3VVN9		no matches	328/1.64	IgG-1.197; IgE-0.951; IgA-1.054	1.27	0.0028	0.1723	-	0.0028	0.0014	0.0153	0.0111
Elongation factor G	Q1GBM0		24.4% identity (52.6% similar) with allergen Phl p 5; E = 0.67; score 0.21	234/1.87	IgG-1.495; IgA-1.246	1.18	0.0029	0.0533	-	0.0029	0.0029	0.0130	0.0101
Enolase	D8FPZ1		50.2% identity (76.3% similar) with allergen Rho m 1; E < 1 ×10^−7^; score 0.75	155//1.2	IgG-1.105; IgA-0.988	1.03	-	-	-	-	-	-	-
Carboxylate-amine ligase	A0A0U3CLC8		31.6% identity (55.7% similar) with alergen Fel d 3; E = 0.59; score: 0.87	200/1.86	IgG- 1.281; IgE-1.296	1.26	0.0048	0.0692	-	0.0072	-	0.0167	0.0095
60 kDa chaperonin (Chaperonin GroEL)	A0A0E3CQW5		26.5% identity (57.6% similar) with alergen Cra g 1.0102; E < 1 ×10^−3^; score: 0.88	212/1.73	IgG-1.237; IgE-0.972; IgA-1.184	0.78	0.0074	0.0423	0.0167	0.0018	-	0.0037	0.0037
UPF0756 membrane protein	B3WE66		44.0% identity with allergen Bla g 1; E = 0.13	108/1.45	IgA-1.048	1.48		0.0327	-	-	-	0.0196	-
30S ribosomal protein S1	Q1GAK0		no matches; score:0.77 to putative allergenes	83/1.27	IgG-1.265; IgE-0.906; IgA-0.958	1.08	0.0025	0.0524	-	-	0.0025	0.0175	0.0125
Lipoprotein	K0MV23		no matchess; core: 0.71 to putative allergenes	133/1.83	IgG-1.045; IgA-1.096	1.09		0.1055	-	-	-	0.0145	-

^a^ UniProt/Trembl database accession number. ^b^ Allergenicity of proteins estimated using Allergen Online database (http://www.allergenonline.org; last update 14 February 2021) using the Full FASTA 36 search algorithm (E-value Cutoff = 1) and with Allergome database (http://www.allergome.org; last update 10 June 2022) using the NCBI blastp algorithm with E-value Cutoff = 1 (cross-reactivity is not likely for proteins with less than 50% identity and similarity and E < 1 × 10^−4^ but for >70% over the entire protein sequence, and E < 1 × 10^−7^ is likely). The Hybrid score for the proteins with the identity to the allergenic sequence based on AlgPred 2.0 web server (https://webs.iiitd.edu.in/raghava/algpred/; last update 17 November 2020) was also provided. ^c^ Hypothetical proinflammatory activity of proteins predicted based on browser ProInflam (http://metagenomics.iiserb.ac.in/proinflam/) (accessed on 1 June 2022) with default settings window length—20 and threshold values—0.7. Values presented as a number of peptides showing inducer potential/score for the strongest inducer. ^d^ The score value of the potential of antibody-specific B-cell epitopes estimated using IgPred web server (https://webs.iiitd.edu.in/raghava/igpred/) (accessed on 31 March 2021) with default settings window length—20 and threshold values—0.9 which scans a protein to identify IgG-, IgA- or IgE- specific B-cell epitopes. ^e^ Hypothetical regulatory activity through ability to induce secretion of IL-10 tested with IL10Pred tool (https://webs.iiitd.edu.in/raghava/il10pred/) (accessed on 31 March 2021) with default settings of window length—9 and threshold values—0.7. Values presented as a score for the strongest inducer. ^f^ Bioactivity of proteins was tested with the BIOPEP tool (https://biochemia.uwm.edu.pl/biopep-uwm/) (accessed on 1 June 2022). Results presented as the A score calculated as the frequency of bioactive fragments occurrence in a protein sequence.

## Data Availability

The data presented in this study are available within the article and supplementary material. The sensitive data (reporting human sera and milk donors) are available on request from the corresponding author.

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
