# Peer review of "The Manifold Bioactivity and Immunoreactivity of Microbial Proteins of Cow and Human Mature Milk in Late Lactation"

_animals, 2022, doi:10.3390/ani12192605_

Round 1
Reviewer 1 Report
Review, paper no. animals-1913084 entitle „The manifold bioactivity and immunoreactivity of microbial proteins of cow and human milk in late lactation”. This is a well-organized study, with sufficient methodology and adequate description of the results. Authors' research has shown interesting relationships. The methodologies used in are in accordance with type of work. The content of microbial proteins in milk was dynamic and varied within species, stage of lactation, and health status of the udder. Generally, the article has some interesting findings which could be worth publication.
Specific comments:
Simple Summary
Correct.
Abstract:
Is sufficiently presented (methods, results, general conclusions).
Generally, nemeration (1), (2), and (3) is unnecessary.
Introduction: The introduction section is sufficient and analytically and adequately covers the need for the study.
Line 75-78. This is a moot point. Should be justified by literature.
Line 90. This a research hypothesis.
Please add aim this work.
Methods: The methodology is sufficiently presented. However, it has a few inaccuracies.
Studies limited a small number of samples of women's milk.
The cows were in the middle lactation period. Please include in all work.
Line 145. Was there a MilkoScan calibration for women's samples.
Results: The results of the study are analytically presented. Figures are adequate explain the findings of the study.
Discussion: The results of study are sufficiently discussed.
Line 816. Could authors define possible limitations of the study? These are just analytical limits.
Conclusion: In conclusion, generalizations are given.
Consider the conclusion of your own research.
Author Response
The authors want to thank to the reviewer for all the valuable comments and suggestions. We clarified all the inconsistencies and shortcomings.
We confirm that the MilcoScan was calibrated to the human milk samples testing.
We also appreciate the comments according the aim and hypothesis clarification. We missed that fact before.
We are also conscious that the small number of human milk samples was a limitation. It is caused because of the idea of purchasing the material from the most homogenous group which still diversed so pooling of samples would be a loose of individual variability. We still have futures planes to broad the scope and involve more participants to this study to compare if the immunoreactivity of microbial proteins will change during early, middle and late phase of lactation.
We also corrected all the recommended places but also performed English correction.
We want to thank for all the effort put into reviews.

Reviewer 2 Report
See attached document

Author Response
Thank you for your effort put into this review and I agree with all the tips and comments. Unfortunately English is not my native language so that's why I've sent the manuscript to external editing. Thank you again for your work and commitment to the manuscript. Yours faithfully Anna Ogrodowczyk
